# Influence of Comprehensive Pre-Anaesthetic Assessment on ASA Classification and Surgical Cancellations in Dogs and Cats: A Retrospective Observational Study

**DOI:** 10.3390/vetsci12070612

**Published:** 2025-06-23

**Authors:** Ariel Cañón Pérez, María De Los Reyes Marti-Scharfhausen Sánchez, Antonio Sevilla Ureba, Eva Zoe Hernández Magaña, Jaime Viscasillas Monteagudo, Agustín Martínez Albiñana, José I. Redondo

**Affiliations:** 1Experimental Surgery Unit, Vall d’Hebron Institut de Recerca (VHIR), Pg. de la Vall d’Hebron 129, Horta-Guinardó, 08035 Barcelona, Spain; 2Anicura Indautxu Hospital, San Mamés Zumarkalea 36–38, 48010 Bilbao, Spain; reyesvet92@gmail.com; 3Urvet Palmones Veterinary Emergency Hospital, Polígono Palmones II, C/Ancla 8, 11379 Cádiz, Spain; amsuvet@gmail.com; 4Departamento de Medicina y Cirugía Animal, Facultad de Veterinaria, Universidad Cardenal Herrera-CEU, CEU Universities, C/Tirant lo Blanc 7, 46115 Valencia, Spain; eva.hernandezmagana@uchceu.es (E.Z.H.M.); nacho@uchceu.es (J.I.R.); 5Anicura Valencia Sur Veterinary Hospital, Av. Picassent 28, 46460 Silla Valencia, Spain; jaimeviscasillas2@gmail.com; 6Anicura Aitana Hospital, C/Xirivella 16, 46920 Valencia, Spain; agustin.martinez.vet@gmail.com

**Keywords:** veterinary anaesthesia, pre-anaesthetic assessment, ASA, Primum Non Nocere, clinical-teaching role, animal safety

## Abstract

Veterinarians routinely check animals before giving them anaesthesia, but it is unclear how much these examinations matter. We reviewed 350 planned procedures in 267 dogs and 83 cats at a Spanish university hospital. Each patient received a thorough history, complete physical examination, and, where indicated, blood tests, X-rays, ECG, and other screens. Hidden health problems were common: the official American Society of Anesthesiologists (ASA) risk grade had to be changed in 7.5% of cases. Even more striking, 16% of the scheduled surgeries were postponed or cancelled, most often because blood tests or imaging uncovered heart or internal-medicine issues that were not obvious on a physical examination alone. About two-thirds of those procedures were successfully rescheduled after treatment or further work-up. These results show that comprehensive pre-anaesthetic assessment can prevent animals from going to the theatre with unrecognised risks, especially when their medical history is sketchy. It supports a targeted but thorough approach to testing.

## 1. Introduction

General anaesthesia is a reversible pharmacological intoxication of the central nervous system that facilitates surgical and diagnostic interventions, causing hypnosis, amnesia, analgesia, and muscular relaxation [1]. Nevertheless, it invariably disturbs physiological homeostasis and exposes patients to measurable risks of morbidity and mortality in both human and veterinary medicine [2,3]. Indeed, despite modern monitoring, peri-anaesthetic death rates remain clinically significant at 0.69% in dogs and 0.63% in cats worldwide [4,5], varying from as low as 0.05% in primary-care settings [6] to 1.35% in tertiary referral populations [7]. 

A structured pre-anaesthetic assessment (PAA) is the cornerstone of clinical anaesthesia. By integrating a comprehensive history, systematic physical examination, and selective complementary diagnostics, PAA establishes a health baseline, identifies occult diseases and comorbidities, and provides critical information for the development of an individualised anaesthetic protocol encompassing dosing, airway management, monitoring, and recovery support, thereby facilitating the provision of appropriate consent information to the patient’s caregivers [8,9,10]. Rigorous PAA correlates with reduced anaesthetic mortality [11], whereas its omission markedly increases intra-operative death [6,12]. Moreover, PAA underpins informed consent, diminishes last-minute cancellations and enhances theatre efficiency [13,14,15].

Risk stratification traditionally relies on the ASA (American Society of Anesthesiologists) Physical Status classification, validated in human and veterinary patients [16,17,18]. However, significant abnormalities frequently emerge without changes in ASA grade, prompting protocol adjustments [19]. For example, Sigrist et al. (2008) demonstrated that routine thoracic radiographs in traumatised dogs and cats often up-staged ASA status and led clinicians to modify ventilation and anaesthetic techniques [20]. Similarly, Joubert (2007) showed that pre-anaesthetic screening in geriatric dogs uncovered subclinical disease in 30% of cases—most commonly neoplasia, renal insufficiency and endocrine disorders—and resulted in postponement or cancellation of anaesthesia in 13% of the patients [21].

The role of blanket complementary testing remains debated: routine laboratory or imaging screens rarely alter management in young, healthy elective patients [22]. However, targeted investigations can reveal actionable pathology in older or systemically unwell animals [23,24]. Clinical context further modulates this balance—patients presented to teaching hospitals or welfare organisations, where histories are often incomplete, may benefit disproportionately from comprehensive screening. In human anaesthesia, deficient pre-operative evaluation has been implicated in 11.6% of intra-operative incidents and nearly 40% of anaesthesia-related deaths [25], underscoring the universal imperative for meticulous pre-anaesthetic assessment.

This study aims to determine how PAA influences perioperative decision-making. Specifically, the study evaluates: (i) patient demographics and reasons for assessment, (ii) the frequency and diagnostic yield of complementary diagnostics, (iii) the rate and causes of ASA physical-status reclassification, and (iv) the proportion of cases in which PAA prompts modification or cancellation of the planned anaesthetic procedure.

We hypothesised that a comprehensive pre-anaesthetic assessment meaningfully alters clinical decision-making, whether by prompting changes to the anaesthetic protocol, additional perioperative precautions, or postponement/cancellation of the procedure.

## 2. Materials and Methods

This retrospective observational study reviewed all PAAs conducted by the Anaesthesiology Service at the CEU Cardenal Herrera University Small Animal Teaching Hospital. Initially, data were systematically collected by anaesthesiology staff in a clinical-academic context. Included were dogs and cats, both privately owned animals and those under the care of animal welfare associations; the latter were predominantly animals awaiting adoption or, in the case of cats, belonged to managed feral colonies. Emergency cases requiring immediate clinical intervention were excluded from the analysis, as PAA documentation in such scenarios was frequently incomplete.

Assessments were documented using a PDF form (see Appendix A: Pre-anaesthetic Assessment Form) accessed via a tablet computer at the point of patient evaluation. The form captured detailed clinical information relevant to the pre-anaesthetic assessment, proposed anaesthetic protocol, and whether informed consent was collected from the animal’s owner or responsible party.

Data Collection

The following information was systematically recorded for each assessment:Administrative and identification data: Case identification number, assessment date, and reason for the pre-anaesthetic evaluation.Patient clinical profile: signalment (species, breed, sex, age, body weight, reproductive status), detailed medical history (both past and current conditions), ongoing medical treatments, previous anaesthetic experiences (including dates, anaesthetic protocols employed, and any recorded complications), and current clinical signs.Physical examination findings: Comprehensive documentation including general appearance and demeanour, body condition, vital parameters (heart rate, respiratory rate, body temperature), mucous membrane colour, hydration status, capillary refill time, thoracic auscultation findings (cardiac and pulmonary), abdominal palpation, lymph node assessment, dermatological examination, oral cavity inspection, neurological evaluation, and locomotor system examination.Initial ASA (classification (ASA-i)): Assigned based exclusively on medical history and physical examination findings, without considering the outcomes of complementary diagnostics (CTS).Complementary diagnostics were requested and performed: Haematologic analysis, serum biochemistry, electrocardiography, radiography, echocardiography, urinalysis, and coagulation profiles, as clinically indicated. Results were categorised as either within normal limits or clinically significant.

Final ASA classification (ASA-f): Re-evaluation of the patient’s ASA status after reviewing the complementary test results. This was conducted by the anaesthesiologist assigned according to the schedule to complete the PAAs. The procedure was not blinded but followed a standardised approach.

Procedure Categorisation

Cases were categorised according to the type of planned surgical or diagnostic intervention as follows:Abdominal procedures (ABDOMINAL): Surgical interventions involving laparotomy (e.g., enterectomy, pyometra surgery, cystotomy, gastrotomy, splenectomy).Diagnostic procedures (DIAGNOSTIC): Anaesthesia administered for diagnostic purposes, such as endoscopy, computed tomography (CT), magnetic resonance imaging (MRI), radiography, and blood sampling.Minor procedures (MINOR): Surgical interventions not requiring the opening of a body cavity, including wound repair, orchiectomy, mastectomy, ophthalmic procedures, and scrotal or perineal hernia repairs.Thoracic procedures (THORACIC): Interventions involving thoracotomy, such as diaphragmatic hernia repair, cardiac or pulmonary surgery, and management of pneumothorax.Trauma procedures (TRAUMA): Orthopaedic or neurological surgeries, including fracture fixation, luxation correction, and hemilaminectomy.

Analysis of Procedure Postponement or Cancellation

In cases where the planned procedure was postponed or cancelled following the PAA, reasons were analysed and systematically classified into seven categories:External reasons (EXTERNAL): Decisions by adoption processes, owners, or welfare groups.Cardiac issues (CARDIO).Dermatological issues (DERMA).Age-related factors, oestrus, or incomplete vaccination status (OTHER).Internal medicine and oncology (IM/ONC).Neurological issues (NEURO).Emergency admission or Trauma-related conditions (EMERG/TRAUMA).

Additionally, the stage during the assessment at which clinically significant findings prompting cancellation were detected was documented, classified as follows:Physical examination and complementary diagnostics (PHE + CTS)Physical examination alone, no CTS required (PHE alone)Continuation of ongoing medical treatment (COT)Complementary diagnostics only (normal physical examination) (CTS alone)Not applicable (NA): decision by owner, welfare organisation, or euthanasia

Finally, data on the subsequent outcome for cancelled procedures (whether the procedure was carried out later) were retrieved from the hospital’s electronic medical records. Any unavailable or incomplete information was classified and recorded as missing data (MD).

## 3. Results

Demographic Data

A total of 350 PAAs were reviewed, comprising 267 dogs (76.3%) and 83 cats (23.7%). Among dogs, 51.3% were female, 42.3% male, with 6.4% lacking sex information (missing data). For cats, 51.8% were female, 33.7% male, and 14.5% MD. The median age (range) and body weight were 6 (0.4–19) years and 14.3 (0.2–58) kg for dogs, and 4 (0.3–13) years and 3.1 (0.9–13.3) kg for cats. Two emergency cases requiring immediate action were excluded from the review due to incomplete data, and were therefore not further examined.

Reasons for Pre-anaesthetic Assessment

Most PAAs were prompted by abdominal and minor procedures, collectively accounting for approximately 75% of the total cases (Figure 1).

Complementary diagnostics

Haematology and serum biochemistry were performed routinely and requested in 81.9–86.1% of assessments in both species. Electrocardiograms were commonly requested for dogs (83.9%) but less frequently for cats (24.1%). Radiographs were performed in most cats (73.5%) and in a significant proportion of dogs (61%). Urinalysis and coagulation testing were requested less frequently.

ASA Classification

For dogs (n = 267), the initial ASA classifications (ASA-i) were: ASA I (30.7%), ASA II (35.6%), ASA III (19.9%), ASA IV (1.9%), and ASA V (0%), with 12% MD. After reviewing complementary test results (ASA-f), 82% of dogs retained their initial classification; 3% (n = 8) decreased by one level; 3% (n = 8) increased by one level, and 12% remained MD. For cats (n = 83), ASA-i distributions were: ASA I (38.6%), ASA II (24.1%), ASA III (19.3%), ASA IV (3.6%), ASA V (0%), with 14.5% MD. Upon comparing ASA-i with ASA-f, 77.1% retained their original classification, 7.2% (n = 6) were increased (half by one level and half by two levels), 1.2% (n = 1) decreased by one level, and 14.5% remained MD. ASA status changed in 7.5% of all cases excluding missing data (Table 1, where the MD was excluded).

Changes in ASA classification represented a refinement of the initial risk assessment, occasionally prompting adjustments in perioperative management and clearer communication with owners regarding the updated risk. An increase in ASA-f was considered indicative of higher anaesthetic risk, potentially prompting adjustments in the perioperative plan and requiring clearer communication with owners regarding the increased risk. Conversely, when the final classification was lower than initially estimated, this reassessment reflected a lower perioperative risk profile and allowed the veterinary team to provide the owner with reassurance at a time that is often associated with stress and uncertainty.

Cancelled Procedures

Fifty-seven procedures (16.2%) were cancelled following pre-anaesthetic assessment, involving 43 dogs (75.4%) and 14 cats (24.6%). Cancellations predominantly involved abdominal (43.9%) and minor (31.6%) procedures. Specific cancelled procedures are detailed in Table 2.

The reasons for cancellation were predominantly related to internal medicine or oncology conditions (IM/ONC), accounting for 50.9% of the cases, followed by cardiological issues (CARDIO), which represented 19.3%. Figure 2 shows cancelled procedures by reason, indicating whether they were later performed, not performed, or if the outcome is unknown due to missing data (MD).

Regarding the outcome of cancelled procedures, 63.2% were later performed, 29.8% were not subsequently carried out, and in 7% the outcome remained unknown (MD). Two animals (3.5% of cancellations) underwent humane euthanasia.

Abnormalities leading to cancellation were first identified through physical examination and complementary diagnostics in 21.1% of cases (n = 12), and through physical examination alone in another 21.1% (n = 12). In one case (1.8%), the procedure was not performed because the ongoing treatment was continued. Most abnormalities (45.6%, n = 26) were detected only after reviewing complementary test results. Finally, 10.5% of cases (n = 6) were classified as ’not applicable’, for those cases in which the cancellation did not result from a medical condition. Results are presented in Table 3.

## 4. Discussion

A comprehensive PAA significantly influenced clinical decision-making, altering the initial anaesthetic approach in approximately one-sixth (16.2%) of cases—a notably higher rate than previously reported veterinary figures, which ranged from 0.8% to 6% [22,24]. Several factors likely contributed to this higher cancellation rate. Firstly, nearly a quarter of our patient population originated from welfare associations or feral cat colonies, which often have incomplete or absent medical histories, increasing the reliance on thorough physical examination and complementary tests (CTS) for detecting underlying disease [26]. Additionally, the dual clinical-teaching role of our university hospital mandated routine CTS (blood work, thoracic radiography, electrocardiography), potentially increasing detection of occult conditions that would not be identified by physical examination alone.

Our data underscore that the physical examination and routine complementary tests substantially impact clinical decisions. Almost half (45.6%) of procedure cancellations were based solely on abnormal CT results, without corresponding physical examination findings, supporting earlier veterinary findings that laboratory and imaging tests can reveal clinically significant abnormalities even in healthy animals [23]. In our study, relevant findings that led to the cancellation of procedures included haematological alterations (such as thrombocytopenia, anaemia, leukopenia, or the detection of *Hepatozoon* spp.) as well as imaging abnormalities, including cardiomegaly, none of which were supported by physical findings. Nevertheless, routine comprehensive screening in all patients remains contentious from a cost-benefit perspective. In human medicine, current guidelines strongly advocate selective, evidence-based testing, reserving comprehensive assessments primarily for older or medically complex patients, as universal testing is neither clinically nor economically justified [27]. A similar, more targeted approach in veterinary medicine, guided by age, ASA classification, and clinical suspicion, might enhance efficiency without compromising safety.

Regarding the ASA physical status classification, our findings align with prior veterinary studies indicating that changes to the ASA grade following PAAs are relatively uncommon (7.5%), but clinically significant when they do occur [19,22]. While ASA reclassification was infrequent, subtle clinical findings—such as previously undetected cardiac murmurs or anaemia—often necessitated modifications to the anaesthetic protocol, underscoring the importance of meticulous clinical examinations beyond ASA scoring alone. Indeed, Louro et al. (2021) similarly reported protocol adjustments in nearly a quarter of canine cases after specialist evaluation, emphasising expert assessment’s enhanced diagnostic accuracy and decision-making confidence [19].

Implementing structured PAA protocols has proven cost-effective by reducing surgical cancellations and improving theatre utilisation in human hospitals, translating into significant economic and resource efficiencies [14,15]. Although our study did not explicitly perform financial analyses, extrapolating these findings suggests that effective PAA practices likely reduce overall costs by avoiding preventable complications, emergency interventions, and wasted theatre resources. Although cost-effectiveness was not evaluated in the present study, future veterinary studies incorporating cost-effectiveness analyses would further clarify the economic implications of various PAA strategies.

A critical limitation of our retrospective study is its inherent reliance on documented records, introducing potential biases related to missing or incomplete data, particularly concerning animals from welfare associations. Such cases frequently lack follow-up, complicating assessment of long-term outcomes. Furthermore, our single-centre academic setting, which routinely conducts extensive diagnostic testing, may limit the generalisability of our findings to private practices or centres with different resource constraints. Another limitation is the inability of this retrospective analysis to establish a direct causal relationship between comprehensive PAA and improved patient outcomes, such as reduced morbidity or mortality. However, extensive epidemiological evidence from both human and veterinary medicine strongly associates structured pre-anaesthetic evaluations with safer anaesthetic outcomes [6,11]. In veterinary practice, documented physical examinations correlate with lower anaesthetic-related mortality, likely due to timely identification and management of risk factors [6]. Although the physical examination protocol is comprehensive, inter-observer agreement was not assessed, which may affect the consistency of findings. Therefore, despite our study’s limitations, existing literature robustly supports the comprehensive PAA’s clinical utility and safety benefits.

Prospective multicentre trials involving structured follow-up and formal outcome assessments are recommended to address these limitations and better quantify PAA’s direct clinical and economic impact. Such studies should focus on perioperative morbidity and mortality and quantify the value of targeted versus routine complementary testing. Additionally, exploring the integration of risk prediction tools beyond ASA classification, such as breed-specific or condition-specific anaesthetic risk indices, may further refine patient stratification and clinical decision-making.

## 5. Conclusions

In conclusion, our findings underscore the importance of comprehensive pre-anaesthetic assessment in veterinary practice, particularly in mixed populations with limited historical information. Although retrospective in design, this study underscores the critical role of meticulous clinical examination and judicious diagnostic testing in reducing risk exposure during anaesthesia. Adopting targeted testing strategies, informed by current veterinary and human evidence, can enhance patient safety, optimise resource use, and uphold the foundational ethical imperative of clinical practice: *Primum Non Nocere*—first, do no harm.

## Figures and Tables

**Figure 1 vetsci-12-00612-f001:**
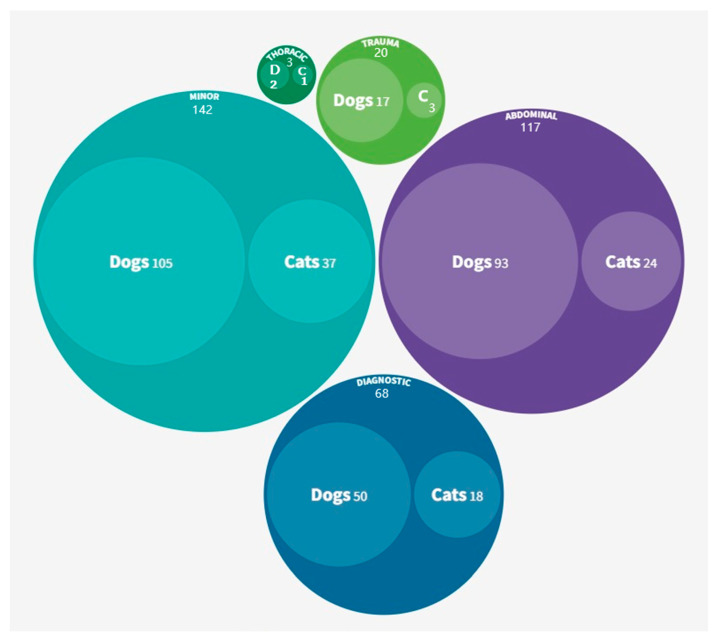
Procedures prompting PAAs categorised by species and overall (n = 350). D, dogs; C, cats.

**Figure 2 vetsci-12-00612-f002:**
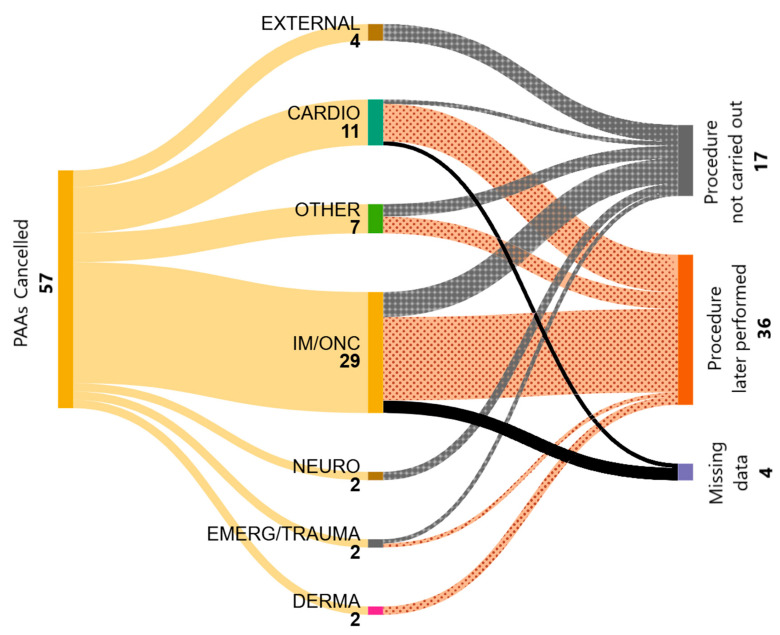
Reasons for procedural cancellations (n = 57) and their subsequent evolution.

**Table 1 vetsci-12-00612-t001:** Change between initial (ASA-i) and final (ASA-f) ASA classifications, **excluding missing data** (n = 306).

Change in ASA (ΔASA)	Cats n (%)	Dogs n (%)	Total n (%)
−2	0 (0)	0 (0)	0 (0)
−1	1 (1.4)	8 (3.4)	9 (2.9)
0 (unchanged)	64 (90.1)	219 (93.0)	283 (92.5)
+1	3 (4.2)	8 (3.4)	11 (3.6)
+2	3 (4.2)	0 (0)	3 (1.0)

**Table 2 vetsci-12-00612-t002:** Specific procedures cancelled after PAAs (n = 57).

Procedure	n	%
Arthrodesis	1	1.8
Biopsy	3	5.3
Endoscopy	2	3.5
Enucleation	1	1.8
Skin tumour excision	2	3.5
Staphylectomy	2	3.5
Bronchoalveolar lavage	1	1.8
Dental cleaning	6	10.5
Ovariohysterectomy (OHE)	18	31.6
OHE + mastectomy	3	5.3
OHE + third eyelid repair	1	1.8
Orchiectomy	5	8.8
Orchiectomy + enucleation	1	1.8
Orchiectomy + rhinoscopy	1	1.8
Osteosynthesis	1	1.8
Retinography	1	1.8
Cranial cruciate ligament repair (CCLR)	4	7.0
CT scan	2	3.5
CT scan + ear canal flush	1	1.8
Missing data	1	1.8
Total	57	100

**Table 3 vetsci-12-00612-t003:** Cancelled procedures categorised by type and source of clinically significant findings. Results expressed as n (%).

Type of Procedure	Cancellations Due to				
	PHE + CTS	PHE Alone	CTS Alone	COT	NA
ABD	7 (12.3%)	5 (8.8%)	8 (14.0%)	1 (1.8%)	4 (7.0%)
DIAG	3 (5.3%)	1 (1.8%)	4 (7.0%)	0 (0.0%)	0 (0.0%)
MINOR	2 (3.5%)	4 (7.0%)	10 (17.5%)	0 (0.0%)	2 (3.5%)
TRAUMA	0 (0.0%)	2 (3.5%)	4 (7.0%)	0 (0.0%)	0 (0.0%)
SUBTOTAL	12 (21.1%)	12 (21.1%)	26 (45.6%)	1 (1.8%)	6 (10.5%)
TOTAL	57 (100%)				

## Data Availability

All relevant data are included in this article.

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
