# Peer review of "Influence of Comprehensive Pre-Anaesthetic Assessment on ASA Classification and Surgical Cancellations in Dogs and Cats: A Retrospective Observational Study"

_vetsci, 2025, doi:10.3390/vetsci12070612_

Round 1

Reviewer 1 Report

Comments and Suggestions for Authors

General Comments:

This is a well-structured and clinically relevant retrospective study that evaluates the role of comprehensive pre-anaesthetic assessments (PAA) in altering ASA classification and surgical decision-making in small-animal patients. The study population—drawn from both privately owned animals and those under welfare organisations—adds further value by reflecting real-world heterogeneity in veterinary practice.

The analysis is thorough and offers actionable findings. In particular, the identification of occult clinical conditions through complementary testing—even in the absence of physical exam abnormalities—reinforces the practical value of PAA beyond theoretical benefit. I support this manuscript after minor revisions.

Specific Comments:

  1. Clarity of ASA Terminology (Section 2.4 and Tables):
    The distinction between ASA-i and ASA-f is helpful and well-defined. However, readers unfamiliar with such retrospective reassessment protocols might benefit from a clearer narrative explanation in the Results section, summarizing how reclassification influenced perioperative planning or altered specific protocols.
  2. Clinical Value of Complementary Tests (Section 3 & 4):
    The finding that nearly half (45.6%) of the cancellations stemmed solely from abnormalities identified via complementary tests—despite normal physical exams—is particularly compelling. I suggest expanding this point slightly in the Discussion, perhaps including an illustrative example (anonymised, of course) of such a case. This would further emphasise the diagnostic yield of these tests and their practical impact on clinical decision-making.
  3. Risk Stratification and Procedural Context (Table 2):
    Given the diversity of procedures (e.g., dental cleaning, OHE, CT), a brief stratification of which types of procedures most commonly led to test-driven cancellations (vs. exam-driven) might further clarify the decision-making patterns clinicians face in day-to-day triage.
  4. Data Presentation Suggestions:
    • Table 1: Add a brief note confirming the exclusion of missing data (n = 306), to reinforce transparency.
    • Figure 2: Consider enhancing readability by refining color contrast or including pattern differentiation for those accessing the paper in greyscale.
  5.  
  6. Minor Stylistic Suggestions:
    • Line 316: “Despite its retrospective nature…” could be revised for smoother flow: e.g., “Although retrospective in design, this study underscores…”
    • Consider using “complementary diagnostics” instead of “complementary tests” for stylistic consistency in certain sections, though both are acceptable

Author Response

Comments 1: Clarity of ASA Terminology (Section 2.4 and Tables):
The distinction between ASA-i and ASA-f is helpful and well-defined. However, readers unfamiliar with such retrospective reassessment protocols might benefit from a clearer narrative explanation in the Results section, summarizing how reclassification influenced perioperative planning or altered specific protocols.

Response 1: Thank you for your valuable suggestion. We have added a paragraph in the Results section providing a narrative explanation of the potential impact that ASA reclassification may have had on perioperative planning and communication with owners.

Comments 2: Clinical Value of Complementary Tests (Section 3 & 4):
The finding that nearly half (45.6%) of the cancellations stemmed solely from abnormalities identified via complementary tests—despite normal physical exams—is particularly compelling. I suggest expanding this point slightly in the Discussion, perhaps including an illustrative example (anonymised, of course) of such a case. This would further emphasise the diagnostic yield of these tests and their practical impact on clinical decision-making

Response 2: We thank you for the observation. In response, we have added specific examples of relevant findings from complementary tests that led to procedure cancellations, to better illustrate their clinical impact. These include haematological alterations (e.g., thrombocytopenia, anaemia, leukopenia, detection of Hepatozoon spp.) and imaging abnormalities (e.g., cardiomegaly) that were not supported by physical examination findings. We believe that including these examples helps readers better appreciate the practical implications of such alterations.

Comments 3: Risk Stratification and Procedural Context (Table 2):
Given the diversity of procedures (e.g., dental cleaning, OHE, CT), a brief stratification of which types of procedures most commonly led to test-driven cancellations (vs. exam-driven) might further clarify the decision-making patterns clinicians face in day-to-day triage.

Response 3: Thank you for your valuable suggestion. In response, we have included Table 3, which presents the information grouped by procedure type and by the source of clinically significant findings (i.e., physical examination, complementary tests, or both). We believe this addition helps to clarify the practical implications of the diagnostic approach and facilitates interpretation of the data.

Comments 4: Data Presentation Suggestions:

  • a) Table 1: Add a brief note confirming the exclusion of missing data (n = 306), to reinforce transparency.

Response 4 a): Thank you for your valuable suggestion regarding transparency about missing data. We have already indicated the exclusion of missing data (n = 306) both in the title of Table 1 (where “excluding missing data (n = 306)” is highlighted in bold) and in the Results section (“Table 1, where the MD was excluded”). We believe these clarifications adequately address your concern. However, we are happy to include an additional brief note if you consider it necessary to further enhance clarity.

  • b) Figure 2: Consider enhancing readability by refining color contrast or including pattern differentiation for those accessing the paper in greyscale.

Response 4b: Thank you for the suggestion. We have modified the figure to ensure optimal visualization in grayscale.

During this revision, we also corrected some percentage values in the results section that were inaccurate in the previous version of the manuscript. These have now been updated accordingly.

Comments 5: Minor Stylistic Suggestions:

  • Line 316: “Despite its retrospective nature…” could be revised for smoother flow: e.g., “Although retrospective in design, this study underscores…”

Response: Thank you for your valuable input. We have revised the text to improve the flow in this section.

  • Consider using “complementary diagnostics” instead of “complementary tests” for stylistic consistency in certain sections, though both are acceptable

Response: Thanks for this suggestion. We have revised the text accordingly and replaced the terms as recommended. 

Reviewer 2 Report

Comments and Suggestions for Authors

Dear Authors,
Thank you for your interesting and well-organized manuscript. The topic is clinically relevant, and your results support the importance of structured pre-anaesthetic assessments. I have suggested a few minor revisions to clarify methodology and improve the interpretation of findings. I believe your study is suitable once these points are addressed.
Best regards,

See below my extended review

Manuscript Title: Influence of comprehensive pre-anaesthetic assessment on ASA classification and surgical cancellations in dogs and cats: A retrospective observational study
Recommendation: Minor Revisions

General Comments:
This is a well-structured and clearly written retrospective observational study that investigates the impact of comprehensive pre-anaesthetic assessment (PAA) on ASA classification and surgical cancellation rates in dogs and cats. The manuscript addresses a clinically relevant topic and contributes to the improvement of perioperative protocols in veterinary practice. The methodology is sound, and the conclusions are mostly supported by the data. A few clarifications and minor improvements would further strengthen the manuscript.

Specific Comments:

  1. Clarify Exclusions: Please state how many emergency cases were excluded and whether any data from these cases were reviewed.
  2. ASA Classification Reassessment: Specify who performed the ASA-f reassessment and whether this was done in a blinded or standardized fashion.
  3. Inter-observer Variability: The physical examination protocol is comprehensive, but there is no mention of inter-observer agreement. Please acknowledge this as a limitation.
  4. Statistical Methods: Briefly expand the description of the statistical analysis. Specify which statistical tests were used to compare ASA-i and ASA-f scores, and whether confidence intervals or p-values were calculated for ASA shifts and cancellation outcomes.
  5. Abstract/Table Consistency: Ensure consistency in the reported values of pain scores or ASA classifications between the abstract and the tables.
  6. Data Visualization: Consider including a figure to visually represent changes in ASA classification or the causes of surgery cancellation.
  7. Discussion Language: Slightly temper the conclusions regarding causality (e.g., “reducing risk exposure” instead of “preventing unnecessary risk exposure”), as this is an observational study.
  8. Cost-effectiveness: Since cost-effectiveness of testing is mentioned in the discussion, please state clearly that this was not evaluated in the present study.

Conclusion:
The topic is clinically important and presented in a clear and well-organized manner. Addressing the few points above will improve the clarity and transparency of the work.

Author Response

Comments 1: Clarify Exclusions: Please state how many emergency cases were excluded and whether any data from these cases were reviewed.

Response 1: Thank you for your comment. We have clarified that two emergency cases were excluded due to incomplete data, which were not further analyzed.

Comments 2: ASA Classification Reassessment: Specify who performed the ASA-f reassessment and whether this was done in a blinded or standardized fashion.

Response 2: Thank you for the observation. We have clarified in the manuscript that the ASA-PS re-evaluation was performed by the anesthesiologist assigned according to the schedule to complete the PAA, and that this was not a blinded procedure.

Comments 3: Inter-observer Variability: The physical examination protocol is comprehensive, but there is no mention of inter-observer agreement. Please acknowledge this as a limitation.

Response 3: Thanks for suggesting the inclusion of this important limitation in our discussion.

Comments 4: Statistical Methods: Briefly expand the description of the statistical analysis. Specify which statistical tests were used to compare ASA-i and ASA-f scores, and whether confidence intervals or p-values were calculated for ASA shifts and cancellation outcomes.

Reponse 4: Thank you for the insightful comment. In the current version of the manuscript, no statistical analysis was performed to compare ASA-i and ASA-f scores, nor were confidence intervals or p-values calculated for ASA shifts or cancellation outcomes, as this section was conceived as a descriptive and exploratory component of the study. However, if the reviewer considers that such an analysis could provide relevant additional insights, we would be pleased to include it in a revised version of the manuscript.

Comments 5: Abstract/Table Consistency: Ensure consistency in the reported values of pain scores or ASA classifications between the abstract and the tables.

Response 5: Thank you for their careful review and for suggesting that we ensure consistency between the data presented in the tables and the text. An error in the percentages of the initial ASA classification in dogs was identified and corrected; the data in the tables were accurate. Please note that in Table 1, missing data were excluded (this was previously mentioned in the table header, but we have now highlighted this in bold for clarity). Additionally, we have explicitly stated in the text where Table 1 is cited that these missing data were excluded, in order to avoid any confusion for the reader when reading the article.

In addition to the requested changes, we have revised and corrected certain percentage values in the results section that were previously inaccurate. These corrections have been incorporated into the updated manuscript.

Comments 6: Data Visualization: Consider including a figure to visually represent changes in ASA classification or the causes of surgery cancellation

Response 6: Thank you very much for your opinion. Based on this comment, which is similar to one made by another reviewer, we have included a new table (Table 3) in the manuscript. This table groups the types of procedures into categories and presents the findings that led to the cancellation of the procedures. Additionally, we have included in the text some examples of abnormalities observed in the complementary diagnostics with the same purpose. We believe these additions clarify and reinforce the importance of PAAs in daily clinical practice.

Comments 7: Discussion Language: Slightly temper the conclusions regarding causality (e.g., “reducing risk exposure” instead of “preventing unnecessary risk exposure”), as this is an observational study.

Response 7: Thanks for this suggestion. We have modified the sentence accordingly.

Comments 8: Cost-effectiveness: Since cost-effectiveness of testing is mentioned in the discussion, please state clearly that this was not evaluated in the present study.

Response 8: Thank you for this observation. We have clarified in the text that cost-effectiveness was not evaluated in the present study.

Comments 9: Conclusion:The topic is clinically important and presented in a clear and well-organized manner. Addressing the few points above will improve the clarity and transparency of the work.

Response 9: We sincerely thank the reviewer for the positive assessment and helpful suggestions. We appreciate your contribution to improving the clarity of the manuscript and will address the points raised accordingly.